# QueryStream: Advancing Streaming Video Understanding with Query-Aware Pruning and Proactive Response

**Kairui Zhang[1], Zhenyu Yang[2,3] Bing Wang[4], Shengsheng Qian[2,3]\*, Changsheng Xu[1,2,3,5]**
[1]ShanghaiTech University, [2]MAIS, Institute of Automation, Chinese Academy of Sciences,
[3]University of Chinese Academy of Sciences, [4]Tianjin University of Technology,
[5]Peng Cheng Laboratory
zhangkr2025@shanghaitech.edu.cn

## Abstract

The increasing demand for real-time interaction in online video scenarios necessitates a new class of efficient streaming video understanding models. However, existing approaches often rely on a query-agnostic "change-is-important" assumption, which conflates visual dynamics with semantic relevance, leading to computational redundancy and mistimed responses. To address this, we propose QueryStream, a novel framework that integrates query-awareness into the core of video processing and response scheduling. QueryStream features two synergistic components: (1) *Query-Aware Differential Pruning* (QDP), a policy that filters the token stream by jointly assessing semantic relevance to the query and temporal novelty against a dynamically smoothed history; and (2) *Relevance-Triggered Active Response* (RTAR), a dual-gated mechanism that schedules responses based on both high query relevance and significant information density. As a lightweight, training-free module, QueryStream achieves state-of-the-art performance on benchmarks such as StreamingBench and OVO-Bench under moderate pruning, and matches full-token baselines while pruning over 70% of visual tokens. Notably, our pruning mechanism generalizes to offline tasks, where it serves as a context-denoising module that benefits long-form video understanding. This work not only reveals the vast semantic redundancy in video streams relative to user intent but also establishes a promising, intent-driven direction for efficient and robust online video understanding. Code is available at: https://github.com/Zhangkr2003/QueryStream.

## 1 Introduction

The paradigm of video understanding is undergoing a fundamental shift from offline, retrospective analysis to online, interactive scenarios, driven by applications such as embodied AI (Duan et al., 2022), autonomous driving (Grigorescu et al., 2020), live event monitoring (Chen et al., 2024a), and real-time interactive editing (Yang et al., 2024). Recent advances in Large Vision-Language Models (LVLMs) (Li et al., 2023a; Dai et al., 2023; Li et al., 2024a; Hurst et al., 2024; Chen et al., 2024b; Bai et al., 2025; Comanici et al., 2025) have catalyzed the development of powerful Video Large Language Models (Video-LLMs) (Maaz et al., 2023; Li et al., 2023b; Ataallah et al., 2024; Zhang et al., 2025a; Wang et al., 2025b; Li et al., 2024a). However, these models are predominantly designed for offline settings, treating video as a static, finite batch of frames. This paradigm is misaligned with the nature of streaming data, which requires processing continuous, unbounded video streams with minimal latency. The sheer volume and inherent temporal redundancy of streaming video render exhaustive, frame-by-frame processing computationally prohibitive and introduce unacceptable response delays. The central challenge, therefore, is to devise mechanisms for intelligent information filtering and proactive response generation, bridging the gap between the power of Video-LLMs and the demands of real-time interaction.

---

\*Corresponding author: shengsheng.qian@nlpr.ia.ac.cn

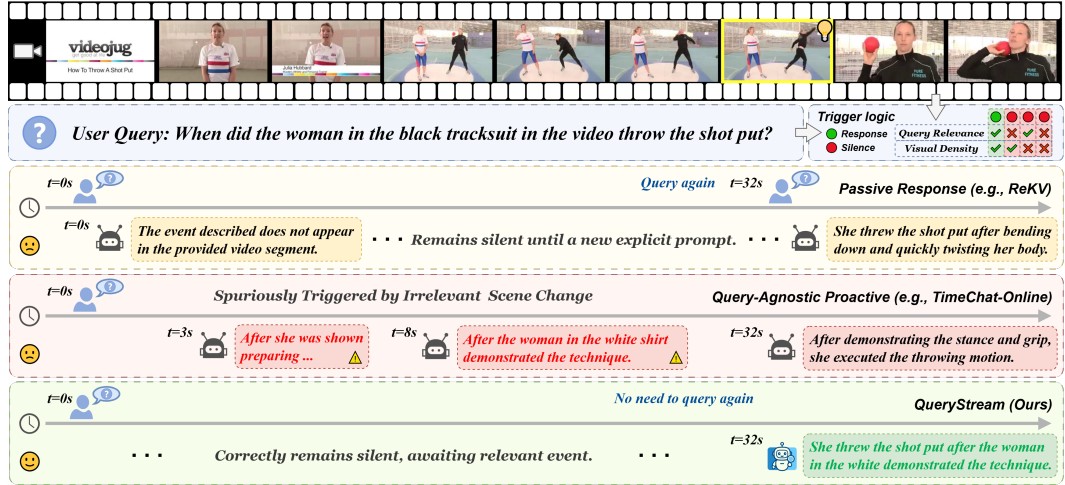

Figure 1: **A comparison of response paradigms in streaming video understanding.** Given a query about a future event, QueryStream remains silent during irrelevant segments and proactively responds only when the queried event occurs, enabling efficient and precise interaction.

To this end, prior work can be broadly categorized into passive and proactive response models. Passive models (Yang et al., 2025b; Di et al., 2025; Huang et al., 2025; Ning et al., 2025) focus on efficient memory management for on-demand querying, but their defining characteristic is that they require a user prompt to trigger a response. In contrast, proactive models (Chen et al., 2024a; Wu et al., 2024b; Wang et al., 2024; Li et al., 2025a; Wang et al., 2025a; Yang et al., 2025c) aim to autonomously determine when to respond. Despite their advanced interactivity, a common drawback of these systems is their reliance on heavily trained, specialized modules for response scheduling, which often compromises their computational efficiency and response accuracy.

More recently, TimeChat-Online (Yao et al., 2025) introduced an elegant approach that leverages visual change detection to concurrently prune redundant tokens and infer opportune moments for response. This *"change-is-important"* philosophy, however, rests on a flawed premise: it conflates raw visual dynamics with true semantic relevance. As illustrated in Figure 1, a model guided by such a principle is thus prone to error. It can be spuriously triggered by visually salient yet semantically irrelevant changes, such as abrupt scene transitions or the actions of actors unrelated to the query, while missing the truly relevant event when it is brief or visually inconspicuous. This disconnect between visual saliency and query-specific importance results in inaccurate responses and wasted computation, highlighting the need for a query-informed paradigm.

To address these limitations, we propose **QueryStream**, a novel framework that instills query-awareness into the core of streaming video understanding for efficient processing and interactive response. As shown in Figure 2, QueryStream is designed to overcome the pitfalls of prior paradigms by redefining information filtering and response scheduling through two synergistic components.

First, we introduce **Query-Aware Differential Pruning (QDP)**, a token pruning strategy that moves beyond naive frame-to-frame comparisons. QDP evaluates the importance of information from two perspectives: semantic relevance to the user's query and temporal novelty. Notably, temporal novelty is determined not against the immediately preceding frame, but against a dynamically smoothed history (DSH) representation of recent frames. This design makes QDP robust to slow visual drifts and transient noise. As a result, a token is preserved only if it satisfies both criteria: (i) it must be semantically relevant to the user's query, and (ii) it must represent a significant temporal deviation from the smoothed historical context. This selective policy ensures that the model's computational focus is directed toward sparse yet meaningful visual dynamics.

Second, we tackle the challenge of timely interaction with a **Relevance-Triggered Active Response (RTAR)** mechanism. Unlike methods that rely on complex learned schedulers (e.g., predicting EOS token) or simple, query-agnostic change detection, RTAR dynamically determines optimal response moments by monitoring two key signals. A response is triggered only when both conditions are

satisfied: (i) the current visual input is highly aligned with the query's semantics, and (ii) there is a significant influx of new, query-relevant information, as reflected by an increase in the token keep rate under the QDP mechanism. This dual-gated policy enables proactive and contextually appropriate interactions.

Our contributions are summarized as follows:

- We propose QueryStream, a novel, training-free framework that establishes a query-centric paradigm for efficient processing and proactive interaction in streaming video understanding. Its modular design allows for seamless integration with off-the-shelf Video-LLMs.

- We introduce Query-Aware Differential Pruning (QDP), a token pruning mechanism that jointly models semantic relevance and temporal novelty using a dynamically smoothed historical context, improving the accuracy and efficiency of token filtering.

- We design the Relevance-Triggered Active Response (RTAR) policy, a dynamic scheduling mechanism that triggers responses based on a dual criterion of semantic relevance and information density, enabling opportune and context-aware interaction.

- Extensive experiments demonstrate that QueryStream achieves state-of-the-art performance on multiple video understanding benchmarks, while substantially improving computational efficiency.

## 2    RELATED WORK

**Streaming Video Understanding.** Streaming video understanding seeks to process continuous video streams in real time for interactive applications. Early approaches can be broadly divided into passive and proactive models. Passive models emphasize efficient memory management for on-demand querying, typically through dynamic KV-caches or memory banks that preserve historical context (Di et al., 2025; Ning et al., 2025; Zhang et al., 2024). While computationally efficient, these models remain purely reactive, generating responses only upon explicit user prompting. Proactive models, in contrast, autonomously decide when to respond, for instance by predicting special EOS tokens (Chen et al., 2024a) or using auxiliary classification heads (Wang et al., 2024). The most relevant work, TimeChat-Online (Yao et al., 2025), introduced an elegant proactive strategy that couples response triggering with visual change detection. However, such proactive methods are inherently query-agnostic: their response policies are governed either by heavily trained, task-specific modules or by the simplistic "change-is-important" heuristic. In contrast, QueryStream introduces a lightweight, logic-driven proactive mechanism (RTAR) that is intrinsically query-aware, thereby enabling more accurate and context-sensitive interactions without additional training.

**Visual Token Pruning.** The redundancy of visual data in videos has motivated substantial research on token pruning. Early approaches compress frames or clips into a fixed number of tokens (Li et al., 2024b; Ren et al., 2024; Xu et al., 2024; Yang et al., 2025a), which fails to adapt to the varying information density of video streams. More advanced methods introduce adaptive pruning strategies, though most remain query-agnostic. A notable example is the Differential Token Drop (DTD) from TimeChat-Online (Yao et al., 2025), which preserves tokens based on inter-frame dissimilarity. While adaptive, its key limitation lies in conflating visual change with semantic importance, a point highlighted in our discussion. Another line of work explores language-guided or query-aware pruning (Song et al., 2024; Zhang et al., 2025b; Li et al., 2025b). However, these methods are largely designed for offline processing and are ill-suited to streaming settings, since they typically require re-processing the entire video history for each new query. Our QDP bridges these paradigms by being adaptive to the evolving video content while remaining sensitive to user intent, and by operating in a streaming-efficient manner that incrementally processes incoming frames without redundant recomputation. Moreover, its incorporation of a dynamically smoothed history for novelty detection enhances robustness beyond simple frame-to-frame comparisons.

## 3    QUERYSTREAM

In this section, we elaborate on the proposed QueryStream framework. QueryStream is a lightweight, plug-and-play module designed to enhance pre-trained Video-Large Language Models

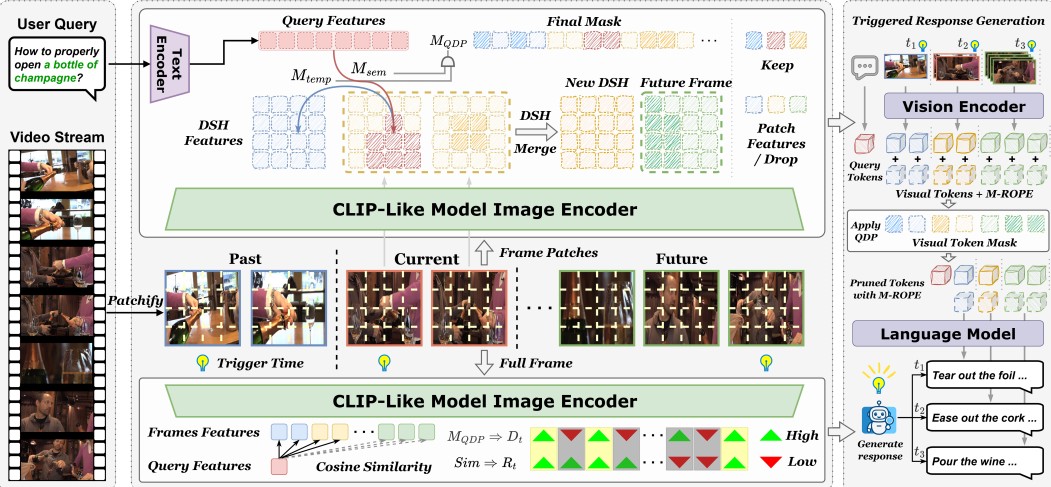

Figure 2: **Overview of the QueryStream framework.** Acting as an intelligent gateway, QueryStream employs Query-Aware Differential Pruning (QDP) to filter tokens by intersecting semantic relevance ($M_{\text{sem}}$) and temporal novelty ($M_{\text{temp}}$). Simultaneously, the Relevance-Triggered Active Response (RTAR) policy monitors relevance ($R_t$) and information density ($D_t$) to trigger the Video-LLM only at optimal moments, ensuring efficient and timely responses.

(Video-LLMs) for online, interactive tasks by instilling query-awareness into their core processing. We begin with a high-level overview of the architecture in Section 3.1, followed by detailed descriptions of its two key technical components: the Query-Aware Differential Pruning (QDP) mechanism in Section 3.2, and the Relevance-Triggered Active Response (RTAR) policy in Section 3.3.

## 3.1 ARCHITECTURAL OVERVIEW

The overall architecture of QueryStream is illustrated in Figure 2. It operates as an intelligent pre-processing gateway that sits between the raw video stream and a backbone Video-LLM (e.g., Qwen2.5-VL (Bai et al., 2025)). Its core philosophy is to align the model's computational focus with the user's intent by establishing a direct interaction between the visual stream and the query's semantics. Given a continuous video stream and a user's query, its primary function is twofold: (i) to judiciously filter out semantically and temporally redundant visual tokens before they reach the computationally expensive Video-LLM, and (ii) to dynamically identify the most opportune moments to trigger a response from the model.

The framework's workflow follows two parallel paths processed by a lightweight, pre-trained vision-language encoder (we use OpenCLIP (Cherti et al., 2023)). The first path, QDP, generates a pruning mask for each frame. The second path, the RTAR policy, analyzes the frame's relevance and information density to decide whether to activate the Video-LLM's decoder. The original, unpruned visual tokens are temporarily held in a memory buffer. Upon receiving a trigger signal from RTAR, the accumulated pruning masks are applied to this buffer of tokens in a just-in-time manner. The resulting sparse token set, along with the query, is then fed into the backbone Video-LLM to generate a timely and contextually grounded response. This architecture ensures that the powerful but resource-intensive Video-LLM is invoked sparingly and purposefully.

## 3.2 QUERY-AWARE DIFFERENTIAL PRUNING

The core of our method is the Query-Aware Differential Pruning (QDP) mechanism, a lightweight module designed to distill a dense visual stream into a sparse, query-relevant token sequence. QDP's philosophy is a stark departure from the conventional "change-is-important" principle. Instead of treating all visual dynamics as equally salient, it employs a dual-criterion sieve that preserves a visual token only if its corresponding patch is (1) semantically aligned with the user's query and (2) temporally novel against a dynamically maintained historical context.

Formally, for a given video stream $V = \{f_1, ..., f_T\}$ and a query $Q$, we first use a lightweight vision-language encoder $\mathcal{E}$ (e.g., OpenCLIP) to extract a feature vector $\mathbf{v}_t^i$ for each patch, yielding a set $\{\mathbf{v}_t^1, ..., \mathbf{v}_t^N\}$ that represents the patch-level features of frame $f_t$, alongside a query embedding $\mathbf{q}$. The pruning process is then governed by two synergistic filtering criteria.

**Semantic Relevance Filtering.** To focus computation on query-pertinent visual information, our first criterion assesses the semantic relevance of each patch. A patch is considered relevant if its feature vector $\mathbf{v}_t^i$ has a similarity to the query embedding $\mathbf{q}$ that exceeds a dynamic, frame-adaptive threshold. This threshold is the average similarity across all features in the current frame, making the filtering robust to varying scene complexities. The semantic mask $M_{\text{sem}}$ is thus defined as:

$$M_{\text{sem}}(t, i) = \mathbb{I}\left(\text{sim}(\mathbf{q}, \mathbf{v}_t^i) > \frac{1}{N}\sum_{j=1}^{N}\text{sim}(\mathbf{q}, \mathbf{v}_t^j)\right),$$

where $\mathbb{I}(\cdot)$ is the indicator function and $\text{sim}(\cdot, \cdot)$ denotes cosine similarity. This ensures that the model's attention is focused exclusively on parts of the scene pertinent to the user's question.

**Temporal Novelty Filtering.** To identify genuine state changes while remaining insensitive to transient noise or gradual environmental shifts, our second criterion evaluates the temporal novelty. We eschew naive frame-to-frame comparisons and instead assess novelty against a **dynamically smoothed history** (DSH). For each patch location $i$, we maintain a historical feature vector $\bar{\mathbf{v}}_{\text{dsh}}^i$. A patch is deemed novel if its feature vector $\mathbf{v}_t^i$ significantly deviates from this established context:

$$M_{\text{temp}}(t, i) = \mathbb{I}\left(\text{sim}(\mathbf{v}_t^i, \bar{\mathbf{v}}_{\text{dsh},t-1}^i) < \tau_{\text{temp}}\right).$$

Following this check, the historical context is updated to integrate the current visual information:

$$\bar{\mathbf{v}}_{\text{dsh},t}^i = \alpha \cdot \mathbf{v}_t^i + (1 - \alpha) \cdot \bar{\mathbf{v}}_{\text{dsh},t-1}^i,$$

where the smoothing factor $\alpha \in [0, 1]$ controls the rate of adaptation. This DSH mechanism provides an adaptive reference, ensuring that only significant departures are flagged as temporally novel.

**Synergistic Pruning Policy.** The final pruning decision is a logical conjunction of these two criteria: a visual token is preserved if and only if its corresponding patch passes both the semantic filter $M_{\text{sem}}(t, i)$ and the temporal filter $M_{\text{temp}}(t, i)$. The final QDP mask is thus computed as:

$$M_{\text{QDP}}(t, i) = M_{\text{sem}}(t, i) \wedge M_{\text{temp}}(t, i).$$

This dual-filter approach ensures the downstream model processes a stream purged of both query-irrelevant and temporally redundant information. Critically, to maintain spatio-temporal integrity, this mask governs the selection of the complete visual tokens. For each preserved patch, both its feature vector and its corresponding Multi-modal Rotary Position Embedding (M-ROPE) are retained. By excising the positional embeddings of discarded patches, we ensure the remaining tokens retain their original and correct {*temporal, height, width*} coordinates. The output of QDP is thus a highly purified, positionally coherent token stream containing only the most salient data for the given query.

### 3.3 RELEVANCE-TRIGGERED ACTIVE RESPONSE

Complementing the QDP's function of determining what to process, our Relevance-Triggered Active Response (RTAR) policy addresses the equally critical question of when to respond. RTAR is a dual-gated mechanism that synchronizes the model's responses with moments of high query-specific information influx. This is achieved by jointly evaluating two complementary conditions—a relevance condition ($R_t$) and a density condition ($D_t$)—before triggering a response.

**Relevance Condition.** The first gate prevents the model from responding during visually active but query-irrelevant segments. To achieve this, it assesses whether the current frame is thematically aligned with the user's query. This condition is met if the holistic relevance of the frame, computed by comparing the query embedding $\mathbf{q}$ with the mean-pooled frame feature vector $\bar{\mathbf{v}}_t$, surpasses a predefined threshold $\tau_{rel}$. Formally:

$$R_t = \mathbb{I}(\text{sim}(\mathbf{q}, \bar{\mathbf{v}}_t) > \tau_{\text{rel}}).$$

**Density Condition.** While relevance is necessary, it is not sufficient. To ensure responses are triggered by new information, our second gate evaluates the frame's information density. We proxy

this by the token keep rate from our QDP mechanism, which naturally quantifies the influx of new, query-relevant information. The density condition is met if this rate exceeds a threshold $\tau_{den}$:

$$D_t = \mathbb{I}\left(\frac{1}{N}\sum_{i=1}^{N} M_{\text{QDP}}(t, i) > \tau_{\text{den}}\right).$$

**Triggering Logic.** A response is generated at timestep $t$ only when both the relevance and density conditions are satisfied, ensuring that the model acts on moments that are both contextually appropriate and informationally rich. The trigger signal $T_t$ is a logical conjunction of the two states:

$$T_t = R_t \wedge D_t.$$

This dual-gated policy prevents two failure modes: it avoids premature responses to irrelevant visual activity while maintaining sensitivity to brief but significant events. By demanding both high relevance and significant information density, RTAR produces responses that are more meaningful, timely, and aligned with the user's interactive intent.

## 4 EXPERIMENTS

### 4.1 EXPERIMENTAL SETUP

**Datasets and Metrics.** Our evaluation comprehensively assesses performance in both online and offline scenarios. For online streaming understanding, we employ two prominent benchmarks: StreamingBench (Lin et al., 2024), a comprehensive benchmark for real-time visual understanding, and OVO-Bench (Niu et al., 2025), which focuses on complex backward tracing and forward active responding capabilities. For offline long-form video understanding, we assess performance on Video-MME (Fu et al., 2025) and LongVideoBench (Wu et al., 2024a). For all question-answering tasks, we adopt accuracy as the primary performance metric. To quantify computational efficiency, we report the Token Keep Rate (%), defined as the percentage of visual tokens retained after pruning.

**Baselines.** We compare QueryStream against a representative set of strong baselines. Our primary comparison is with TimeChat-Online (Yao et al., 2025), as it represents the most relevant prior work based on query-agnostic differential pruning. We also include other leading streaming Video-LLMs such as Flash-VStream(Zhang et al., 2024), VideoLLM-online (Chen et al., 2024a) and Dispider (Qian et al., 2025). To provide a performance reference, we further compare against the original Qwen2.5VL-7B (Bai et al., 2025), which processes the full, unpruned stream of visual tokens.

**Implementation Details.** Our QueryStream model is implemented by replacing the query-agnostic pruning module in TimeChat-Online-7B with our proposed query-aware mechanisms. Additionally, to evaluate the zero-shot generalization of our pruning strategy, we also integrate the QDP module into the base Qwen2.5VL-7B model. For the feature extraction that underpins our pruning decisions, we utilize the publicly available OpenCLIP-ViT-L/14 (Cherti et al., 2023) as our lightweight vision-language encoder $\mathcal{E}$. The DSH smoothing factor is set to $\alpha = 0.1$. The thresholds for temporal novelty ($\tau_{\text{temp}}$), relevance ($\tau_{\text{rel}}$), and density ($\tau_{\text{den}}$) are determined on a small held-out validation set (see Appendix A.6) and are applied consistently across all experiments. Unless specified otherwise, our model processes video streams at 1 FPS. Crucially, QueryStream requires no additional fine-tuning; all results are achieved in a zero-shot, plug-and-play manner, underscoring its adaptability and ease of integration. All experiments are conducted on a single NVIDIA 80 GB A800 GPU.

### 4.2 MAIN RESULTS ON STREAMING VIDEO BENCHMARKS

**Performance on StreamingBench.** The results on StreamingBench, detailed in Table 1, clearly demonstrate the superiority of QueryStream's intent-driven filtering over the query-agnostic approach of TimeChat-Online. At a moderate token keep rate of 57.2%, QueryStream achieves an overall score of 75.32, surpassing TimeChat-Online (74.32 with a 55.8% keep rate) by a significant 1.0-point margin. Notably, this score nearly matches the performance of the full-token TimeChat-Online baseline (75.36), demonstrating substantial computational savings with negligible performance impact. The advantage of query-awareness becomes even more pronounced under aggressive pruning. With a highly efficient token keep rate of just 29.6%, QueryStream's score of 74.04

Table 1: **Performance comparison on StreamingBench.** The table benchmarks a comprehensive suite of models, including proprietary, open-source offline, and online Video-LLMs. A key comparison is drawn between our proposed QueryStream and the strong TimeChat-Online baseline under varying token keep rates. The results highlight QueryStream's ability to achieve state-of-the-art performance while operating with significantly fewer visual tokens.

| Model | #Frames | Keep Rate(%) | OP | CR | CS | ATP | EU | TR | PR | SU | ACP | CT | All |
|---|---|---|---|---|---|---|---|---|---|---|---|---|---|
| Human | - | - | 89.47 | 92.00 | 93.60 | 91.47 | 95.65 | 92.52 | 88.00 | 88.75 | 89.74 | 91.30 | 91.46 |
| **Proprietary MLLMs** | | | | | | | | | | | | | |
| Gemini 1.5 pro | 1 fps | - | 79.02 | 80.47 | 83.54 | 79.67 | 80.00 | 84.74 | 77.78 | 64.23 | 71.95 | 48.70 | 75.69 |
| GPT-4o | 64 | - | 77.11 | 80.47 | 83.91 | 76.47 | 70.19 | 83.80 | 66.67 | 62.19 | 69.12 | 49.22 | 73.28 |
| Claude 3.5 Sonnet | 20 | - | 73.33 | 80.47 | 84.09 | 82.02 | 75.39 | 79.53 | 61.11 | 61.79 | 69.32 | 43.09 | 72.44 |
| **Open-source Offline VideoLLMs** | | | | | | | | | | | | | |
| Video-LLaMA2-7B | 32 | - | 55.86 | 55.47 | 57.41 | 58.17 | 52.80 | 43.61 | 39.81 | 42.68 | 45.61 | 35.23 | 49.52 |
| VILA-1.5-8B | 14 | - | 53.68 | 49.22 | 70.98 | 56.86 | 53.42 | 53.89 | 54.63 | 48.78 | 50.14 | 17.62 | 52.32 |
| Video-CCAM-14B | 96 | - | 56.40 | 57.81 | 65.30 | 62.75 | 64.60 | 51.40 | 42.59 | 47.97 | 49.58 | 31.61 | 53.96 |
| LongVA-7B | 128 | - | 70.03 | 63.28 | 61.20 | 70.92 | 62.73 | 59.50 | 61.11 | 53.66 | 54.67 | 34.72 | 59.96 |
| InternVL-V2-8B | 16 | - | 68.12 | 60.94 | 69.40 | 77.12 | 67.70 | 62.93 | 59.26 | 53.25 | 54.96 | 56.48 | 63.72 |
| Kangaroo-7B | 64 | - | 71.12 | 84.38 | 70.66 | 73.20 | 67.08 | 61.68 | 56.48 | 55.69 | 62.04 | 38.86 | 64.60 |
| LLaVA-NeXT-Video-32B | 64 | - | 78.20 | 70.31 | 73.82 | 76.80 | 63.35 | 69.78 | 57.41 | 56.10 | 64.31 | 38.86 | 66.96 |
| MiniCPM-V-2.6-8B | 32 | - | 71.93 | 71.09 | 77.92 | 75.82 | 64.60 | 65.73 | 70.37 | 56.10 | 62.32 | 53.37 | 67.44 |
| LLaVA-OneVision-7B | 32 | - | 80.38 | 74.22 | 76.03 | 80.72 | 72.67 | 71.65 | 67.59 | 65.45 | 65.72 | 45.08 | 71.12 |
| Qwen2.5-VL-7B | 1 fps | - | 78.32 | 80.47 | 78.86 | 80.45 | 76.73 | 78.50 | 79.63 | 63.41 | 66.19 | 53.19 | 73.68 |
| **Open-source Online VideoLLMs** | | | | | | | | | | | | | |
| Flash-VStream-7B | - | - | 25.89 | 43.57 | 24.91 | 23.87 | 27.33 | 13.08 | 18.52 | 25.20 | 23.87 | 48.70 | 23.23 |
| VideoLLM-online-8B | 2 fps | - | 39.07 | 40.06 | 34.49 | 31.05 | 45.96 | 32.40 | 31.48 | 34.16 | 42.49 | 27.89 | 35.99 |
| Dispider-7B | 1 fps | - | 74.92 | 75.53 | 74.10 | 73.08 | 74.44 | 59.92 | 76.14 | 62.91 | 62.16 | 45.80 | 67.63 |
| TimeChat-Online-7B | 1 fps | 55.8% | 81.03 | 83.59 | 78.55 | 81.09 | 76.73 | 80.37 | 75.93 | 63.82 | 68.47 | 47.87 | 74.32 |
| TimeChat-Online-7B | 1 fps | 33.0% | 81.03 | 82.03 | 77.60 | 82.37 | 73.58 | 79.13 | 77.78 | 62.20 | 66.48 | 39.89 | 72.96 |
| TimeChat-Online-7B | 1 fps | 100% | 80.22 | 82.03 | 79.50 | 83.33 | 76.10 | 78.50 | 78.70 | 64.63 | 69.60 | 57.98 | 75.36 |
| **QueryStream-7B** | 1 fps | 57.2% | 82.38 | 84.38 | 79.18 | 82.37 | 77.99 | 81.31 | 78.70 | 65.04 | 69.32 | 47.34 | **75.32** |
| **QueryStream-7B** | 1 fps | 29.6% | 82.11 | 83.59 | 78.23 | 82.69 | 75.47 | 80.06 | 79.63 | 63.01 | 67.90 | 42.55 | **74.04** |

still outperforms TimeChat-Online (72.96 with a 33.0% keep rate) by 1.08 points, despite processing even fewer tokens. This consistently superior performance validates that query-aware pruning acts as an effective context-denoising mechanism. By filtering out semantically irrelevant visual noise, QueryStream provides the model with a cleaner context. This benefit is particularly evident in reasoning-heavy sub-tasks; for instance, at the 60% keep rate level, it outperforms its counterpart on Causal Reasoning (CR) and Text-Rich Understanding (TR) by 0.79 and 0.94 points, respectively.

**Performance on OVO-Bench.** On OVO-Bench, a benchmark designed to test complex reasoning, QueryStream's advantages are further pronounced (Table 2). With a token keep rate of 52.9%, our model establishes a new state-of-the-art score of 49.4 among all online models, surpassing even the full-token TimeChat-Online (46.7) by a significant 2.7-point margin. This superior performance is not achieved at the cost of efficiency; on the contrary, under an aggressive pruning regime that keeps only 20.0% of tokens, QueryStream (47.5) still maintains a substantial performance lead over both the compressed (47.6 at 55.4% keep rate) and full-token (46.7) versions of its query-agnostic counterpart. A closer inspection reveals that this performance gain is consistent across all three major categories, with the most significant improvements observed in the more challenging *Backward Tracing* and *Forward Active Responding* tasks. This suggests that our intent-driven filtering provides a more robust context for complex temporal reasoning.

## 4.3 PERFORMANCE ON OFFLINE LONG-VIDEO TASKS

To assess the generalization of our query-aware pruning, we evaluate its efficacy on offline long-video benchmarks, with results in Table 3 showing compelling performance across both scenarios.

**Results on VideoMME.** On VideoMME, our query-aware approach demonstrates a clear advantage over query-agnostic methods and even full-token processing. First, when QDP is applied as a zero-shot module to the base Qwen2.5-VL-7B, it achieves a score of 63.6 with a 52.4% token keep rate, outperforming its full-token counterpart (63.2). This counter-intuitive finding—achieving superior performance with less data—validates our hypothesis that QDP acts as an effective context-denoising mechanism. The advantage is particularly pronounced on the challenging "long" subset, where our method surpasses the baseline by a substantial 2.2-point margin (52.6 vs. 50.4). Our full QueryStream model further confirms this superiority, scoring 63.8 and outperforming the comparable TimeChat-Online configuration (63.3) at a similar efficiency level.

Table 2: **Evaluation results on OVO-Bench.** OVO-Bench comprises three challenging categories: (i) *Real-Time Visual Perception*, (ii) *Backward Tracing*, and (iii) *Forward Active Responding*. Our proposed QueryStream is benchmarked against a comprehensive suite of models. The results highlight its state-of-the-art performance among online models, demonstrating robust capabilities on complex temporal reasoning tasks while operating with significantly fewer visual tokens.

| Model | #Frames | Real-Time Visual Perception | | | | | | | Backward Tracing | | | | Forward Active Responding | | | | Overall |
|---|---|---|---|---|---|---|---|---|---|---|---|---|---|---|---|---|---|
| | | OCR | ACR | ATR | STU | FPD | OJR | Avg. | EPM | ASI | HLD | Avg. | REC | SSR | CRR | Avg. | Avg. |
| Human Agents | - | 94.0 | 92.6 | 94.8 | 92.7 | 91.1 | 94.0 | 93.2 | 92.6 | 93.0 | 91.4 | 92.3 | 95.5 | 89.7 | 93.6 | 92.9 | 92.8 |
| *Proprietary Multimodal Models* | | | | | | | | | | | | | | | | | |
| Gemini 1.5 Pro | 1fps | 87.3 | 67.0 | 80.2 | 54.5 | 68.3 | 67.4 | 70.8 | 68.6 | 75.7 | 52.7 | 62.3 | 35.5 | 74.2 | 61.7 | 57.2 | 65.3 |
| GPT-4o | 64 | 69.1 | 65.1 | 65.5 | 50.0 | 68.3 | 63.7 | 63.6 | 49.8 | 71.0 | 55.4 | 58.7 | 27.6 | 73.2 | 59.4 | 53.4 | 58.6 |
| *Open-source Offline VideoLLMs* | | | | | | | | | | | | | | | | | |
| LLaVA-NeXT-Video-7B | 64 | 69.8 | 59.6 | 66.4 | 50.6 | 72.3 | 61.4 | 63.3 | 51.2 | 64.2 | 9.7 | 41.7 | 34.1 | 67.6 | 60.8 | 54.2 | 53.1 |
| LLaVA-OneVision-7B | 64 | 67.1 | 58.7 | 69.8 | 49.4 | 71.3 | 60.3 | 62.8 | 52.5 | 58.8 | 23.7 | 45.0 | 24.8 | 66.9 | 60.8 | 50.9 | 52.9 |
| Qwen2-VL-7B | 64 | 69.1 | 53.2 | 63.8 | 50.6 | 66.3 | 60.9 | 60.7 | 44.4 | 66.9 | 34.4 | 48.6 | 30.1 | 65.7 | 50.8 | 48.9 | 52.7 |
| InternVL-V2-8B | 64 | 68.5 | 58.7 | 69.0 | 44.9 | 67.3 | 56.0 | 60.7 | 43.1 | 61.5 | 27.4 | 44.0 | 25.8 | 57.6 | 52.9 | 45.4 | 50.1 |
| LongVU-7B | 1fps | 55.7 | 49.5 | 59.5 | 48.3 | 68.3 | 63.0 | 57.4 | 43.1 | 66.2 | 9.1 | 39.5 | 16.6 | 69.0 | 60.0 | 48.5 | 48.5 |
| *Open-source Online Video-LLMs* | | | | | | | | | | | | | | | | | |
| Flash-VStream-7B | 1fps | 25.5 | 32.1 | 29.3 | 33.7 | 28.9 | 28.8 | 29.9 | 36.4 | 33.8 | 5.9 | 25.4 | 5.4 | 67.3 | 60.0 | 44.2 | 33.2 |
| VideoLLM-online-8B | 2fps | 8.1 | 23.9 | 12.1 | 14.0 | 45.5 | 21.2 | 20.8 | 22.2 | 18.8 | 12.2 | 17.7 | - | - | - | - | - |
| TimeChat-Online-7B | 1fps (55.4%) | 74.5 | 48.6 | 68.1 | 48.3 | 69.3 | 59.8 | 61.4 | 56.9 | 64.9 | 11.8 | 44.5 | 31.8 | 38.5 | 40.0 | 36.8 | 47.6 |
| TimeChat-Online-7B | 1fps (15.2%) | 69.8 | 48.6 | 64.7 | 44.9 | 68.3 | 55.4 | 58.6 | 53.9 | 62.8 | 9.1 | 42.0 | 32.5 | 36.5 | 40.0 | 36.4 | 45.6 |
| TimeChat-Online-7B | 1fps (100%) | 75.2 | 46.8 | 70.7 | 47.8 | 69.3 | 61.4 | 61.9 | 55.9 | 59.5 | 9.7 | 41.7 | 31.6 | 38.5 | 40.0 | 36.7 | 46.7 |
| **QueryStream-7B** | 1fps (52.9%) | 75.2 | 49.5 | 69.8 | 50.0 | 71.3 | 62.5 | 63.1 | 56.9 | 65.5 | 12.4 | 44.9 | 35.5 | 43.3 | 41.7 | 40.2 | **49.4** |
| **QueryStream-7B** | 1fps (20.0%) | 74.5 | 47.7 | 70.7 | 46.6 | 71.3 | 57.6 | 61.4 | 54.2 | 63.5 | 8.6 | 42.1 | 33.2 | 43.1 | 40.8 | 39.0 | **47.5** |

Table 3: **Results on offline long-video benchmarks.** We report accuracy on LongVideoBench and VideoMME (w/o subtitles). Our QDP module is evaluated as a zero-shot plug-in on Qwen2.5-VL (w/ QDP), and we also report the performance of QueryStream.

| Model | #Frames | LongVideoBench | VideoMME | |
|---|---|---|---|---|
| | | | overall | long |
| *Video Length* | - | *8 sec∼60 min* | *1∼60 min* | *30∼60 min* |
| *Open-Source Offline VideoLLMs* | | | | |
| LLaMA-VID-7B | 1fps | - | - | - |
| MovieChat-7B | 2048 | - | 38.2 | 33.4 |
| LLaVA-Next-Video-7B | 32 | 43.5 | 46.6 | - |
| VideoChat2-7B | 16 | 39.3 | 39.5 | 33.2 |
| LongVA-7B | 128 | - | 52.6 | 46.2 |
| Kangaroo-7B | 64 | 54.2 | 56.0 | 46.6 |
| Video-CCAM-14B | 96 | - | 53.2 | 46.7 |
| VideoXL-7B | 128 | - | 55.5 | 49.2 |
| Qwen2.5-VL-7B | 1fps (100%) | 61.5 | 63.2 | 50.4 |
| Qwen2.5-VL-7B w/ DTD | 1fps (53.8%) | 61.6 | 63.4 | 51.9 |
| **Qwen2.5-VL-7B w/ QDP** | 1fps (52.4%) | **61.9** | 63.6 | 52.6 |
| *Open-source Online VideoLLMs* | | | | |
| Dispider-7B | 1fps | - | 57.2 | - |
| VideoChat-Online-8B | 2fps | - | 52.8 | 44.9 |
| TimeChat-Online-7B | 1fps (100%) | 55.4 | 62.4 | 48.4 |
| TimeChat-Online-7B | 1fps (53.7%) | 57.1 | 63.3 | 52.4 |
| TimeChat-Online-7B | 1fps (15.0%) | 57.7 | 62.5 | 49.2 |
| **QueryStream-7B** | 1fps (52.4%) | 57.3 | **63.8** | **52.9** |
| **QueryStream-7B** | 1fps (16.6%) | **58.0** | 63.2 | 49.8 |

Table 4: **Ablation of QDP components on StreamingBench.** We analyze the individual and synergistic effects of semantic and visual pruning criteria.

| Pruning Method | Keep(%) | Score (All) |
|---|---|---|
| No Pruning (Baseline) | 100.0 | 75.36 |
| + Visual Pruning Only | 63.4 | 74.76 |
| + Semantic Pruning Only | 61.7 | 74.52 |
| **QueryStream (Full QDP)** | **57.2** | **75.32** |

Table 5: **Ablation of the RTAR triggering policy.** Results on OVO-Bench's *Forward Active Responding* tasks, comparing accuracy (Acc.) with the score-based metric (Score) that rewards both accuracy and timeliness.

| Triggering Method | Acc. (Avg.) | Score (Avg.) |
|---|---|---|
| *Baseline:* | | |
| TimeChat-Online (Density-Only) | 36.8 | 29.5 |
| *QueryStream Variants:* | | |
| Relevance-Only Trigger | **40.3** | 30.2 |
| **Full RTAR (Ours)** | 40.2 | **34.6** |

**Results on LongVideoBench.** The benefits of our approach are further confirmed on LongVideoBench. At a moderate token keep rate of 52.4%, QueryStream (57.3) already outperforms the TimeChat-Online baseline (57.1). More compellingly, under an aggressive pruning regime that retains only 16.6% of tokens, QueryStream's performance not only remains highly competitive but improves to 58.0. This suggests that for very long videos with substantial redundancy, aggressive, query-aware filtering is not just beneficial for efficiency but can be critical for enhancing model focus and accuracy. Collectively, these findings show that our query-aware approach is not just a streaming optimization but a robust paradigm for efficient long-video understanding.

## 4.4 ABLATION STUDIES

To dissect the architecture of QueryStream and validate our key design choices, we conduct a series of detailed ablation studies. We aim to quantify the individual and synergistic contributions of the components within our QDP and RTAR mechanisms.

**Effectiveness of QDP Components.** To understand the interplay between the semantic and temporal filters in QDP, we conduct an ablation study with results shown in Table 4. The analysis reveals a powerful synergistic effect. Applying either the Temporal Pruning Only or Semantic Pruning Only filter in isolation leads to a minor but noticeable performance degradation compared to the no-pruning baseline. This suggests that while each filter reduces token count, their individual criteria are not precise enough to fully separate signal from noise. Remarkably, our full QDP method, which forms the intersection of these two criteria, resolves this trade-off. It achieves the highest efficiency with the lowest token keep rate of 57.2% while restoring performance to a level virtually identical to the full-token baseline. These results demonstrate that the two filters are complementary. Their combination yields a stricter and more precise policy that removes noise each filter alone would retain. By preserving only tokens that are both semantically relevant and temporally novel, QDP delivers a purified context that sharpens model focus and maximizes accuracy at minimal cost.

**Impact of the DSH Smoothing Factor.** We conduct a sensitivity analysis on the DSH smoothing factor $\alpha$ to evaluate the benefit of a smoothed historical context over naive frame-to-frame comparisons. As shown in Figure 3, $\alpha$ governs a clear trade-off between efficiency and performance on OVO-Bench. A large $\alpha = 1.0$ makes the model overly sensitive to noise, leading to poor performance, while decreasing $\alpha$ stabilizes the historical context and improves accuracy. Performance peaks at $\alpha = 0.1$, where the model achieves the best balance between score and token efficiency. Further reducing $\alpha$ results in an overly long memory, slowing adaptation and degrading performance. These results confirm the importance of a smoothed, medium-term history in DSH.

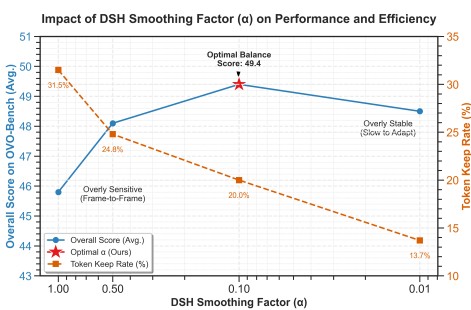

Figure 3: Effect of the DSH smoothing factor ($\alpha$) on performance (Overall Score) and efficiency (Token Drop Rate) on OVO-Bench.

**Analysis of the RTAR Triggering Policy.** To demonstrate the superiority of our dual-gated RTAR policy ($R_t \wedge D_t$), we conduct an ablation on OVO-Bench's *Forward Active Responding* tasks, with results in Table 5. The analysis compares raw accuracy (Acc.) with a timeliness-aware metric (Score). The Density-Only trigger, mirroring TimeChat-Online's method, yields the lowest score (29.5) because it often activates on irrelevant dynamic events. In contrast, the Relevance-Only trigger achieves the highest accuracy (40.3) but is penalized for timeliness, resulting in a low score of 30.2, since it generates redundant responses for static yet relevant scenes. Our full RTAR policy strikes the optimal balance by attaining near-peak accuracy (40.2) together with a score of 34.6, which is 4.4 points higher than the next best variant. This result confirms that the synergy of the relevance and density gates is crucial for producing responses that are both contextually appropriate and informationally novel and timely. Detailed calculation methods are provided in Appendix A.4.

## 5   CONCLUSION

In this paper, we introduced QueryStream, a novel query-centric framework for efficient and interactive streaming video understanding that departs from conventional query-agnostic approaches. QueryStream establishes this paradigm through two synergistic, training-free components: Query-Aware Differential Pruning (QDP) for precise token filtering and Relevance-Triggered Active Response (RTAR) for timely interaction. Extensive experiments demonstrate that QueryStream achieves state-of-the-art performance on streaming benchmarks while processing significantly fewer tokens, and generalizes effectively to offline tasks as a context-denoising module. Overall, this work highlights the semantic redundancy of video streams relative to user intent and suggests a more efficient, intent-driven direction for streaming video understanding.

ACKNOWLEDGMENTS

This work is supported by the National Key Research and Development Program of China (No.2023YFC3310700), the Beijing Natural Science Foundation (JQ23018, L252032), and the National Natural Science Foundation of China (No.U25B2076, 62522218, 62276257).

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

# A APPENDIX

## A.1 STATEMENT ON THE USE OF LARGE LANGUAGE MODELS

In line with the conference policy, we disclose that Large Language Models (LLMs) were used solely as writing aids. Their involvement was limited to improving grammar, refining sentence structure, and enhancing readability. All scientific contributions, including the development of ideas, methodology, experiments, and conclusions, were made exclusively by the authors, who take full responsibility for the content of this paper.

## A.2 ETHICS STATEMENT

We have read and adhere to the ICLR Code of Ethics. Our research is conducted solely on publicly available benchmarks for video understanding, and all datasets are used in accordance with their licenses. Our framework leverages large pre-trained models (e.g., Qwen2.5-VL, TimeChat-Online, OpenCLIP), which, like others of this type, may reflect limitations of their training data. While our method does not directly address such issues, it does not introduce additional risks. The intended use of QueryStream is to improve the efficiency and responsiveness of interactive video understanding systems, such as assistive technologies or monitoring tools. A positive ethical aspect is its contribution to sustainability: by reducing processed tokens, our method lowers computational cost and energy consumption. We declare no competing interests.

## A.3 REPRODUCIBILITY STATEMENT

We have made every effort to ensure the reproducibility of our work. The source code for QueryStream, including the implementation of our Query-Aware Differential Pruning and Relevance-Triggered Active Response methods, will be made publicly available upon publication. Our framework is built upon publicly available models. For the base Video-LLM, our experiments utilize both Qwen2.5-VL-7B and TimeChat-Online-7B. The feature encoder used is OpenCLIP-ViT-L/14. All relevant citations for these models are provided in the main text. All datasets used in our experiments, including StreamingBench, OVO-Bench, VideoMME, and LongVideoBench, are standard and publicly available benchmarks. Critical hyperparameters and detailed experimental settings are documented in Section 4.1. Furthermore, the Appendix provides a comprehensive description of our simulated evaluation protocol for the active response tasks (Appendix A.4) and an analysis of our key component choices (Appendix A.5), further aiding the reproducibility of our results.

## A.4 SIMULATED EVALUATION PROTOCOL FOR THE RTAR ABLATION STUDY

This section details the simulated evaluation protocol used for the RTAR ablation study presented in Table 5, which focuses on the *Forward Active Responding* category of OVO-Bench. As the official online evaluation code for OVO-Bench and the real-time inference code for TimeChat-Online were not publicly available at the time of our experiments, we devised a simulated evaluation methodology designed to fairly approximate the benchmark's intended real-time assessment.

Our simulation proceeds as follows. For a given video, we first let both models process the entire stream to identify all potential response trigger points according to their respective mechanisms:

- For QueryStream, a trigger is registered at any timestep $t$ where our RTAR policy ($T_t$) fires.
- For TimeChat-Online, we simulate its density-based trigger by identifying timesteps where its token keep rate (the inverse of the drop rate) is significantly higher than a baseline threshold, indicating a moment of high visual change.

From the sequence of trigger timestamps generated by each model, we identify distinct event intervals. For each interval, we select the first timestamp as the definitive response point, $t'_m$. This simulates a model making its first response upon detecting a new, relevant event and prevents duplicate evaluations for a single continuous event.

Finally, to generate the actual response $R_{m'}$, we feed only the video frames up to and including the trigger timestamp $t'_m$ into the respective model and perform inference. The resulting response $R_{m'}$

is then compared against the ground-truth answer $A_m$ to calculate correctness using the function $F(R_{m'}, A_m)$. Based on this, we compute the two final metrics:

- **Accuracy (Acc.)**: The average correctness across all responses, providing a direct measure of response quality.

$$\text{Acc} = \frac{1}{N} \sum_{i=1}^{N} F(R_{m'}, A_m)$$

- **Score**: A metric that jointly rewards accuracy and timeliness. It penalizes the temporal deviation of the response from the ideal moment $t_m$ using an absolute difference $|t'_m - t_m|$. This design ensures that **both premature and delayed responses are penalized**, encouraging the model to act precisely when sufficient evidence becomes available.

$$\text{Score} = \sum_{i=1}^{N} F(R_{m'}, A_m) \cdot 2^{-|t'_m - t_m| \cdot p}$$

This simulated protocol ensures a fair and consistent comparison, as each model's performance is evaluated based on the context available only up to the point where its own internal logic decided to act. While the initial identification of trigger points leverages the full video stream—a necessary simplification to enable offline evaluation—the subsequent response generation strictly adheres to temporal constraints, thus closely approximating a real-world online scenario.

## A.5 ANALYSIS OF COMPONENT SELECTION FOR QUERYSTREAM

The modular design of our QueryStream framework allows the QDP mechanism to be integrated with various feature encoders and base Video-LLMs. This section details the empirical analysis conducted to justify our final selection of OpenCLIP-ViT-L/14 as the feature encoder and Qwen2.5-VL-7B as the base model.

For an efficient yet representative analysis, we first created a validation subset by randomly sampling approximately 10% of the data (around 250 samples) from each task category in OVO-Bench. We then evaluated the performance of our QDP module when paired with different combinations of popular OpenCLIP variants and state-of-the-art Video-LLMs. The results, measured by the OVO-Bench Overall Score, are presented in Table 6.

Table 6: **Component selection for QueryStream.** Performance (OVO-Bench Overall Score) of our QDP module with different feature encoders and base Video-LLMs on a validation subset. Our final choice, highlighted in gray, balances performance, efficiency, and fairness.

| Feature Encoder | Qwen2.5-VL-7B | InternVL2.5-8B | InternVideo2.5-8B |
|---|---|---|---|
| OpenCLIP-ViT-B/32 | 47.2 | 46.9 | 47.8 |
| **OpenCLIP-ViT-L/14** | **48.5** | **48.2** | **49.0** |
| OpenCLIP-ViT-H/14 | 48.9 | 48.4 | 49.2 |

The results yield two key insights. First, for any given Video-LLM, using a more powerful CLIP encoder (from ViT-B to ViT-H) generally leads to improved performance. This confirms the importance of high-quality feature representations for effective pruning. However, this performance gain comes at the cost of significant computational overhead and latency, particularly with larger encoders like ViT-H/14, which is prohibitive for a real-time system. Second, the results demonstrate the versatility of our QDP module, which successfully enhances the performance of various leading Video-LLMs, underscoring its plug-and-play nature.

Based on this analysis, our final component selection was guided by three core principles: (i) **Performance**, the combination must deliver strong results on the target benchmark; (ii) **Efficiency**, the feature encoder must be lightweight enough to support real-time operation; and (iii) **Fair Comparison**, the chosen base Video-LLM should align with our primary baseline for an equitable comparison.

Consequently, OpenCLIP-ViT-L/14 was chosen as it offers the best trade-off between feature quality and computational efficiency. Qwen2.5-VL-7B was selected as the base model because it is not only a strong and representative open-source model but, critically, it also serves as the foundation for our main baseline, TimeChat-Online. This choice ensures that our observed performance gains can be more directly attributed to our proposed query-aware mechanisms rather than differences in the underlying model architectures.

## A.6 Hyperparameter Selection

The logic-based gates in our QueryStream framework rely on three key thresholds: the temporal novelty threshold ($\tau_{\text{temp}}$) for QDP, and the relevance ($\tau_{\text{rel}}$) and density ($\tau_{\text{den}}$) thresholds for RTAR. These values were determined empirically on the same held-out validation set described in Appendix A.5. Our goal was to find a robust set of parameters that balances performance and efficiency.

Table 7: Impact of the temporal novelty threshold ($\tau_{\text{temp}}$) on token keep rate and overall performance on the OVO-Bench validation subset. The selected value is highlighted.

Table 8: Impact of RTAR thresholds ($\tau_{\text{rel}}$, $\tau_{\text{den}}$) on the average Score on the OVO-Bench *Forward Active Responding* validation subset. The selected values are highlighted.

| $\tau_{\text{temp}}$ | Keep Rate (%) | Overall Score |
|---|---|---|
| 0.75 | 21.6 | 46.5 |
| 0.85 | 35.1 | 48.2 |
| **0.90** | 52.9 | 49.4 |
| 0.95 | 68.3 | 49.1 |
| 0.98 | 78.5 | 49.0 |

| $\tau_{\text{rel}}$ | $\tau_{\text{den}}$ | Average Score |
|---|---|---|
| 0.50 | 0.15 | 33.1 |
| 0.70 | 0.15 | 33.8 |
| 0.60 | 0.10 | 32.5 |
| 0.60 | 0.20 | 33.2 |
| **0.60** | 0.15 | 34.6 |

**Determining the Temporal Novelty Threshold ($\tau_{\text{temp}}$).** The threshold $\tau_{\text{temp}}$ directly controls the filtering aggressiveness of our QDP module. A lower value leads to more aggressive pruning (lower keep rate). We performed a sweep over a range of values for $\tau_{\text{temp}}$ and evaluated its impact on the overall performance (Score) on the OVO-Bench validation subset.

As shown in Table 7, a value of $\tau_{\text{temp}} = 0.90$ achieved the best overall score. While a more aggressive threshold of 0.85 offered a lower keep rate, it began to degrade performance, suggesting that critical temporal information was being erroneously pruned. Conversely, a more lenient threshold of 0.95 retained more tokens without providing a commensurate performance gain, indicating that it allowed too much redundancy. Therefore, we selected $\tau_{\text{temp}} = 0.90$ for all our experiments as it strikes the optimal balance between efficiency and accuracy.

**Determining the RTAR Thresholds ($\tau_{\text{rel}}$ and $\tau_{\text{den}}$).** The RTAR thresholds govern the active response policy and were tuned to maximize the timeliness-aware Score metric on the *Forward Active Responding* tasks from the same OVO-Bench validation subset. We performed a grid search to analyze the interplay between the relevance and density gates.

The results, summarized in Table 8, indicate that a combination of $\tau_{\text{rel}} = 0.60$ and $\tau_{\text{den}} = 0.15$ yields the highest score. Deviating from these values hurts performance: a lower relevance threshold ($\tau_{\text{rel}} = 0.50$) caused erroneous triggers on irrelevant scenes, while a higher one ($\tau_{\text{rel}} = 0.70$) missed some valid response opportunities. Similarly, a lower density threshold ($\tau_{\text{den}} = 0.10$) was overly sensitive to minor visual noise, whereas a higher one ($\tau_{\text{den}} = 0.20$) failed to trigger on more subtle events. The chosen values represent the most robust configuration for triggering responses that are both accurate and timely.

## A.7 Case Study

To provide an intuitive understanding of QueryStream's operational advantages, we present a single, illustrative case study in Figure 4. The video is specifically engineered to probe two common failure modes of query-agnostic systems: false trigger from irrelevant visual shocks and miss trigger from subtle, relevant events. To simulate these challenges, we insert several black frames to represent a transient shock and include a visually subtle but query-relevant action. The figure visualizes the models' token drop patterns and their resulting responses, offering a direct comparison between QueryStream and TimeChat-Online.

**False Trigger: Robustness to Irrelevant Shocks.** The first challenge tests the models' robustness to irrelevant motion. As depicted, the query-agnostic TimeChat-Online falters when confronted with the inserted black frames. Guided by its "change-is-important" philosophy, it perceives the abrupt transition as a major visual event, causing its token drop rate to plummet. This leads to a spurious trigger, prompting an erroneous and unhelpful response about the screen going black. In stark contrast, QueryStream's QDP mechanism recognizes that the black frames are semantically

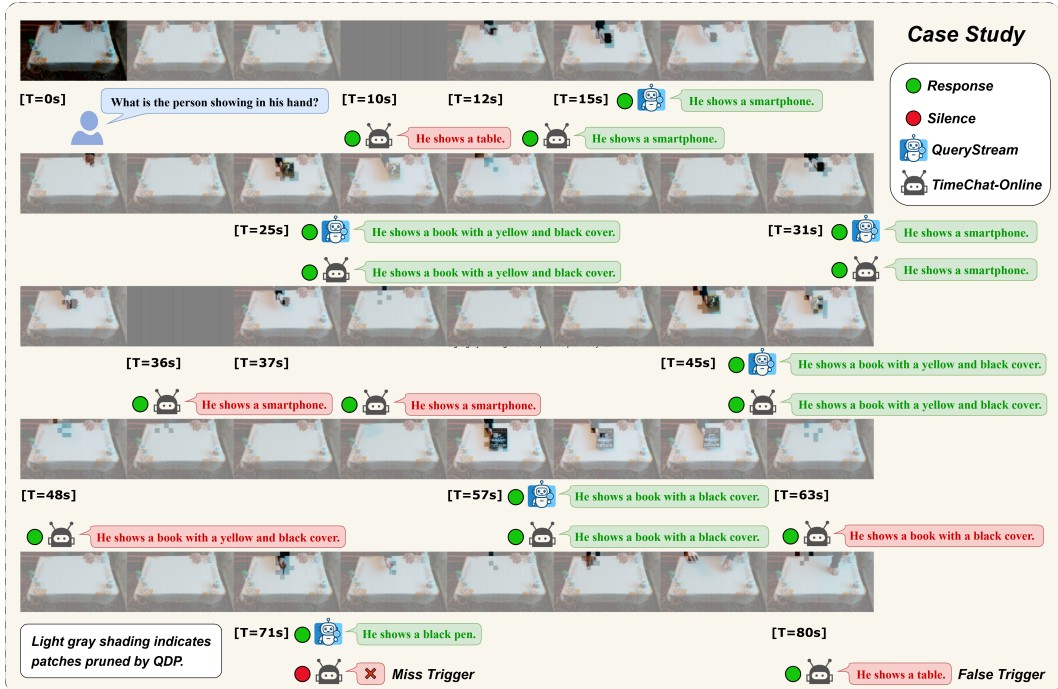

Figure 4: **Qualitative comparison of QueryStream and TimeChat-Online in a challenging case study.**

irrelevant to the user's query. Consequently, it continues to prune these tokens, maintaining a high drop rate and correctly remaining silent, thus demonstrating its ability to distinguish visual dynamics from semantic importance.

**Miss Trigger: Sensitivity to Subtle Events.** The case study also includes a slow but crucial action relevant to the query, testing the models' sensitivity. TimeChat-Online's myopic frame-to-frame comparison fails to register the small inter-frame differences of this subtle action, thus missing the event entirely and resulting in a missed trigger. QueryStream, however, excels in this scenario. Its DSH-based novelty detector registers the persistent, cumulative deviation from the established historical norm. Because this slow change is also highly relevant to the query, both of RTAR's gates are satisfied, leading to a timely and accurate response. This case highlights the synergistic power of our dynamically smoothed history and query-aware triggering, enabling a far more nuanced and intelligent interaction.

## A.8 LIMITATIONS

Despite its strong performance and efficiency, QueryStream has several limitations that highlight promising avenues for future research. Firstly, the efficacy of our QDP mechanism is fundamentally bound by the representational quality of the pre-trained OpenCLIP encoder. Its inherent constraints in discerning fine-grained details or abstract relationships may challenge the pruning precision in semantically nuanced scenarios, potentially causing critical but subtle events to be missed. Secondly, the current framework is designed around a single, static user query and does not explicitly handle dynamic conversational contexts where user intent might evolve over multiple turns. Extending the model to manage a continuously updated query state or dialogue history is a crucial next step. Finally, our mechanism relies on a set of fixed hyperparameters ($\alpha$, $\tau_{\text{temp}}$, $\tau_{\text{rel}}$, $\tau_{\text{den}}$) for its logic-based gates. While effective, these static thresholds may not be universally optimal. Future work could explore adaptive thresholding mechanisms or even learned, soft-gating policies to enable more nuanced, data-driven decision-making and enhance the model's robustness across diverse video domains and query types.

