# OpenReview forum: "QueryStream: Advancing Streaming Video Understanding with Query-Aware Pruning and Proactive Response"
_ICLR.cc/2026/Conference — ICLR 2026 Poster_

### Official Review · Reviewer_ioxx · 2025-10-23

**Soundness:** 4
**Presentation:** 4
**Contribution:** 3
**Rating:** 6
**Confidence:** 4

**Summary:**

This paper introduces QueryStream, a training-free, plug-and-play framework that improves streaming video understanding by aligning video token processing and response timing with the semantic intent of a user’s query. Traditional streaming models follow a “change-is-important” rule that triggers responses based on visual variation, leading to irrelevant activations and inefficiency. QStream aims to solve this.

**Strengths:**

* Shifts from the long-standing “visual-change” heuristic to a query-centric paradigm for efficient video understanding.
* Requires only a small OpenCLIP encoder and simple logic-based gating, making it suitable for real-time applications.
* Works with any existing Video-LLM (e.g., Qwen2.5-VL) without retraining.
* The combination of semantic relevance and temporal novelty is intuitive, interpretable, and effective.
* Sets new SOTA on StreamingBench and OVO-Bench, surpassing trained baselines with far fewer tokens.

**Weaknesses:**

* Despite strong framing, the core mechanisms (cosine similarity, thresholding, dual gating) are algorithmically simple and largely heuristic.
* Several thresholds may need tuning for each setup; no adaptive or learned strategy is explored.
* Tests are limited to mid-scale models (7B) and benchmarks where scalability to larger multimodal LLMs or real-time deployment remains unclear.
* While efficiency is discussed, actual wall-clock or energy improvements are not quantified.
* The novelty primarily lies in combining existing ideas (query relevance + temporal novelty + dual gating) rather than developing a fundamentally new principle.
* Table 1 bolds the results for QStream but there are baselines with better performance even though using more frames.
* Table 3 could benefit from stronger baselines.

Minor comments:
* Authors use \citet instead of \citep.
* No need to repeat QDP many times.

**Questions:**

* Why do you need the semantic relevante filtering and the relevance condition? Aren't they doing the same? It seems unclear to me and not well explained the difference. I'd provide more details.
* Could you report latency and throughput improvements to validate real-time claims?
* How easy would it be to implement an adaptive threshold tuning approach?
* Could you evaluate on other models besides QwenVL2.5 (eg. VideoLLaMA3, InternVL3)?

**Details Of Ethics Concerns:**

I do not identify any significant ethical issues in this paper. The method operates on publicly available video datasets commonly used in the community, and there is no indication of privacy violations, harmful content generation, or misuse potential beyond standard concerns in visual understanding research. Therefore, I do not see any ethical concerns requiring further attention.

---

> ### Author Response · Authors · 2025-11-20
>
> Many thanks to your valuable comments and questions, which help us a lot to improve our work. We address your questions as follows.
>
> ---
>
> > [W1, W5] The core mechanisms are heuristic and simple... with novelty lying in the combination of existing ideas rather than a new principle.
>
> Thank you for your feedback on our methodology. Our framework is founded on the clear principle of **query-awareness**, which posits that information filtering should be driven by user intent. This is a significant departure from the query-agnostic "change-is-important" paradigm used in prior work (e.g., TimeChat-Online). Similarly, it goes beyond passive, query-sensitive models (e.g., ReKV) because it enables autonomous responsiveness and performs pruning at the image level, rather than relying on cached retrieval. A key aspect of our contribution is a general, **training-free**, plug-and-play module that requires only a standard CLIP-like model to **transform** a Video-LLM into an effective streaming video understanding system with **proactive** responsiveness.
>
> > [W2, Q3] It uses manual per-setup threshold tuning... without exploring how easily an adaptive approach could be implemented.
>
> We thank you for this comment regarding our design methodology and hyperparameters. We acknowledge this aspect was discussed in our limitations section, but we believe this configurability is a practical advantage. The use of adjustable hyperparameters provides significant **operational flexibility**, allowing the system's behavior to be tailored to specific application needs. For instance, a surveillance video might require the retention of more fine-grained information, necessitating a higher sensitivity to temporal changes. In contrast, a narrative video summary might only require focusing on major scene shifts, thus benefiting from a lower sensitivity to minor variations. Our framework allows for the configuration of appropriate thresholds to meet these differing requirements, providing explicit control points for task-specific adaptation. This stands in contrast to an end-to-end model where such sensitivity levels are implicitly learned and fixed after training, offering less adaptability at inference time. Furthermore, although these thresholds are static in our experiments, our method achieved excellent performance across multiple benchmarks, which demonstrates its versatility.
>
> Regarding the implementation of an adaptive approach, we see this as a promising direction for future work with several viable paths. For example, one could implement a Test-Time Adaptation (TTA) strategy where thresholds are dynamically adjusted based on real-time statistics of the video stream. A more sophisticated solution could employ Reinforcement Learning (RL) to train an agent that learns an optimal policy for adjusting the thresholds to maximize a long-term reward balancing accuracy, timeliness, and efficiency. Our current logic-based framework provides an interpretable testbed for developing and evaluating such advanced adaptive strategies.
>
>
> > [W3] Tests are limited to mid-scale models (7B) and benchmarks where scalability to larger multimodal LLMs or real-time deployment remains unclear.
>
> Regarding your concern about the performance of our method on larger models, we want to clarify that our framework is independent of model type and size; the actual efficiency improvement is directly related to the scale of the backbone Video-LLM model. This is because our QDP and RTAR methods are implemented based on CLIP-like models, which are independent of the Video-LLM used during inference. Our method can save computational resources by pruning and removing **over 70%** of the tokens; the performance improvement will be even greater for larger, more computationally expensive models.

---

> > ### Author Response · Authors · 2025-11-20
> >
> > > [W4, Q2] Latency and throughput improvements are not quantified.
> >
> > We agree that this is an important omission. As stated in our response to Reviewer 6ZdJ, we added new experiments to measure the end-to-end inference speed of the model with and without pruning. We also quantified the time required to process a single frame using our lightweight CLIP encoder and the full Video-LLM. The results are shown in the table below.
> >
> > | Model | Token Keep Rate (%) | Decision Latency (ms/frame) | Inference Latency (ms/sample) |
> > | :--- | :--- | :--- | :--- |
> > | Qwen2.5VL (Full Token) | 100% | - | 3968 |
> > | TimeChat-Online | 38.00% | 70.57ms | 2310 (×1.72 speedup) |
> > | **QueryStream (Ours)** | **38.18%** | **47.01ms (×1.50 speedup)** | **2129 (×1.86 speedup)** |
> >
> > The table clearly demonstrates two key advantages of our approach. First, our decision-making process is **1.5x faster** per frame than TimeChat-Online's, confirming the benefit of our lightweight module. Second, this initial efficiency contributes to a superior **end-to-end speedup of 1.86x** over the full-token baseline, outperforming the baseline's own pruning method at a similar token keep rate. We will add this comprehensive efficiency analysis to the revised version of the paper to empirically verify the performance improvement.
> >
> >
> > > [W6] Table 1 bolds the results for QStream but there are baselines with better performance even though using more frames.
> >
> > We appreciate your keen observation. We would like to clarify that the bolded sections in Table 1 are intended to highlight the best performance achieved with **similar token keep rates**; therefore, the comparison is not with the full token (100%) model. Specifically, our QueryStream (57.2% keep rate) scored 75.32, almost identical to the full token baseline model's 75.36, while significantly outperforming the query-independent compressed model TimeChat-Online (55.8% keep rate, 74.32 score). The same applies to keep rates around 30%. We will elaborate on this more clearly in the paper.
> >
> > > [W7, Q4] The evaluation could be strengthened by adding stronger baselines to Table 3 and testing on other models beyond just Qwen2.5-VL.
> >
> > We thank you for these valuable suggestions on strengthening our experimental validation. We agree that evaluating on a broader range of models and baselines is an important step. While our framework is designed to be plug-and-play, integrating it with a new Video-LLM requires some engineering work to properly handle the pruned tokens. Additionally, conducting a thorough evaluation across these new configurations will also take some time. We are conducting these additional experiments and will incorporate the results into the final version of the paper.
> >
> >
> > > [Q1] The difference between semantic relevance filtering and the relevance condition is unclear... they seem redundant and require a better explanation.
> >
> > We'd be happy to explain the difference. The two mechanisms are complementary, not redundant, as they operate on different modules, at different granularities, and for distinct purposes. In short, Remantic Relevance Filtering determines which patches of video frames are retained, while Relevance Condition determines when to respond.
> >
> > Specifically，**Semantic Relevance Filtering** is a component of our QDP module and operates at the patch-level. For each incoming video frame, this filter calculates the relevance between the user's query and each patch of the video frame to obtain the mask. **Relevance Condition** is a component of our RTAR module and operates at the frame-level. This condition calculates the relevance between the user's query and the entire video frame, using it as one of the conditions for whether a response is needed for that frame.
> >
> > > [Minor comments] Writing problems
> > >
> > Many thanks for your careful proofreading and for these helpful stylistic suggestions. You are right that we misused the citation format, and we will correct it throughout the manuscript. Additionally, we will revise the paper to reduce the repetition of acronyms to improve the overall readability. We appreciate you pointing these details out to us.
> >
> > ---
> >
> > Thank you once again for your thoughtful and constructive reviews. We have carefully considered all your comments and will incorporate these changes into the revised version. We hope our response and planned modifications will resolve your issues, and we look forward to further discussions with you.

---

> ### Comment · Reviewer_ioxx · 2025-11-25
> **Official Comment by Reviewer ioxx**
>
> Thanks authors for addressing my main concerns. I'm thus raising my score.

---

### Official Review · Reviewer_UzVR · 2025-10-30

**Soundness:** 3
**Presentation:** 3
**Contribution:** 3
**Rating:** 6
**Confidence:** 3

**Summary:**

This paper addresses the inefficiencies of existing streaming video understanding models, where many rely on a flawed "change-is-important" principle that confuses visual dynamics with semantic relevance, leading to computational waste and interaction errors. To solve this, the authors propose QueryStream, a lightweight, training-free framework that embeds query-awareness into video processing and response scheduling. It integrates two main contributions: 1. Query-Aware Differential Pruning, which filters visual tokens by jointly evaluating semantic relevance to the user’s query and temporal novelty. 2. Relevance-Triggered Active Response, which dynamically determines optimal response moments by monitoring two key signals.

**Strengths:**

1. Query-Awareness Solves Core Inefficiencies: Unlike query-agnostic methods, it avoids spurious triggers from irrelevant visual changes and captures subtle but query-relevant events, reducing computational waste and improving response accuracy.
2. Requires no additional fine-tuning, enabling plug-and-play integration with pre-trained Video-LLMs.
3. Clear presentation and illustrations, making the paper easy to follow.
4. Sufficient experiments, establishing the robustness of the proposed method.

**Weaknesses:**

1. QDP’s performance is limited by the representational capacity of the pretrained encoder. It may struggle with fine-grained details or abstract semantic relationships, leading to missed subtle events in nuanced scenarios.
2. The framework assumes a single, fixed user query and cannot handle dynamic conversational contexts where user intent evolves over multiple turns
3. RTAR and QDP rely on static thresholds tuned on a validation set. These thresholds may not be optimal across all video domains, reducing the adaptability.

**Questions:**

See weaknesses.

---

> ### Author Response · Authors · 2025-11-20
>
> Many thanks to your valuable comments and questions, which help us a lot to improve our work. We address your questions as follows.
>
> ---
>
> > [W1] QDP's performance is limited by its encoder... struggling with fine-grained details and potentially missing subtle events.
>
> We thank the reviewer for this comment. Regarding the choice of the CLIP-like model, it is important to note that it operates independently of the Video-LLM. While adding this module improves performance, we have observed that beyond a certain representational capacity, the gains from using a more powerful encoder are limited. This is evidenced by our new tests on SigLIP, as mentioned in our response to Reviewer 6ZdJ. We acknowledge that for highly abstract queries, the encoder's limitations can indeed pose a challenge, and we have discussed this in our limitations section as a direction for future work. However, based on the current experimental results, our method has achieved excellent performance across a rich set of benchmarks, which is sufficient to demonstrate the effectiveness of our approach.
>
>
> > [W2] The framework assumes a single, fixed query... and cannot handle dynamic, multi-turn conversations.
>
> The point raised regarding conversational contexts is indeed important. While we considered it a limitation in our paper, we wish to clarify that our framework's design inherently supports dynamic, multi-turn interactions. The query embedding that guides our QDP and RTAR modules can be updated at any point. When a new user query arrives in a conversation, we simply replace the existing embedding with one generated from the new query. This process is highly efficient as it does not require re-processing past video segments; the system immediately adapts its filtering logic to the new user intent for all subsequent frames. Therefore, handling evolving queries is not a fundamental architectural challenge but a direct application of our flexible design. We will revise the manuscript to better reflect these dynamic capabilities.
>
> Furthermore, for scenarios involving a single persistent query that applies to multiple events, our system is designed to provide proactive responses for each relevant occurrence, as demonstrated in our case study in Appendix A.5 (Figure 4).
>
>
> > [W3] RTAR and QDP use static thresholds... which may not be optimal across all domains, limiting adaptability.
>
> We thank the reviewer for this comment regarding our design methodology and hyperparameters. Our framework is founded on the clear principle of **query-awareness**, which posits that information filtering in streaming video should be guided by user intent, rather than by query-agnostic visual dynamics. We acknowledge this aspect was discussed in our limitations section, but we believe this configurability is a practical advantage. The use of adjustable hyperparameters provides significant operational flexibility, allowing the system's behavior to be tailored to specific application needs. For instance, a surveillance video might require the retention of more fine-grained information, necessitating a higher sensitivity to temporal changes. In contrast, a narrative video summary might only require focusing on major scene shifts, thus benefiting from a lower sensitivity to minor variations. Our framework allows for the configuration of appropriate thresholds to meet these differing requirements, providing explicit control points for task-specific adaptation. This stands in contrast to an end-to-end model where such sensitivity levels are implicitly learned and fixed after training, offering less adaptability at inference time.
>
> Furthermore, although these thresholds are static in our experiments, our method achieved excellent performance across multiple benchmarks, which demonstrates its versatility.
>
> ---
>
> Thank you once again for your thoughtful and constructive reviews. We have carefully considered all your comments and will incorporate these changes into the revised version. We hope our response and planned modifications will resolve your issues, and we look forward to further discussions with you.

---

> > ### Comment · Reviewer_UzVR · 2025-11-25
> >
> > Thanks for the response, which addressed most of my concerns.

---

### Official Review · Reviewer_vYGP · 2025-11-01

**Soundness:** 3
**Presentation:** 3
**Contribution:** 3
**Rating:** 4
**Confidence:** 3

**Summary:**

The paper introduces QueryStream, a novel framework designed to enhance the efficiency and interactivity of streaming video understanding models by incorporating query-awareness into the core processing loop. This work addresses the limitations of existing approaches, which typically rely on a flawed, query-agnostic “change-is-important” principle that conflates raw visual dynamics with true semantic relevance, leading to computational waste and interaction errors in real-time online video scenarios.

**Strengths:**

1. Query-Aware Differential Pruning (QDP): QDP is a novel token pruning mechanism that is unique because it employs a dual criterion that jointly assesses semantic relevance to the user's query and temporal novelty. Previous token pruning methods were often query-agnostic.
2. Dynamically Smoothed History (DSH): Within QDP, temporal novelty is assessed not against the immediately preceding frame, but against a Dynamically Smoothed History (DSH). This adaptive historical context ensures the pruning is robust to transient noise and slow visual drifts.
3. Relevance-Triggered Active Response (RTAR): RTAR is a novel, logic-driven dual-gated mechanism that schedules responses based on a specific confluence of two signals: high query relevance and significant information density. This departs from prior reactive (passive) models or proactive models relying on heavily trained, specialized modules.
4. Lightweight and Training-Free: QueryStream is designed as a lightweight, training-free module that operates as an intelligent pre-processing gateway, allowing for seamless, zero-shot integration with existing off-the-shelf Video-LLMs.

**Weaknesses:**

1. The effectiveness of the core Query-Aware Differential Pruning (QDP) mechanism is "fundamentally bound by the representational quality of the pre-trained OpenCLIP encoder". The authors utilized OpenCLIP-ViT-L/14, which was chosen as a pragmatic trade-off between feature quality and real-time computational efficiency. However, the inherent constraints of this encoder in discerning "fine-grained details or abstract relationships" may challenge the "pruning precision in semantically nuanced scenarios". This reliance risks causing critical but subtle events to be erroneously pruned and missed.
2. The logic-based gating mechanisms (QDP and RTAR) rely on a "set of fixed hyperparameters". These values were determined empirically via sweeps and grid searches on a held-out validation set to find an optimal balance. However, these static thresholds, while effective in the evaluated benchmarks, "may not be universally optimal" and could reduce the model's robustness across diverse video domains, streaming qualities, or different types of user queries.
3. The crucial ablation study for the Relevance-Triggered Active Response (RTAR) policy (Table 5), which demonstrates its superiority in timeliness-aware scoring, was conducted using a simulated evaluation protocol. This simulation was necessary because the official real-time evaluation infrastructure for the OVO-Bench benchmark and the TimeChat-Online code were unavailable at the time of experiments. While the simulation aimed to be fair by adhering to temporal constraints during response generation, the initial identification of trigger points leveraged the full video stream.

**Questions:**

See weakness above, please.

---

> ### Author Response · Authors · 2025-11-20
>
> Many thanks to your valuable comments and questions, which help us a lot to improve our work. We address your questions as follows.
>
> ---
>
> > [W1] QDP's performance is capped by its OpenCLIP encoder... which may miss subtle events due to a lack of fine-grained understanding.
>
> We thank the reviewer for this comment. Regarding the choice of the CLIP-like model, it is important to note that it operates independently of the Video-LLM. While adding this module improves performance, we have observed that beyond a certain representational capacity, the gains from using a more powerful encoder are limited. This is evidenced by our new tests on SigLIP, as mentioned in our response to Reviewer 6ZdJ. We acknowledge that for highly abstract queries, the encoder's limitations can indeed pose a challenge, and we have discussed this in our limitations section as a direction for future work. However, based on the current experimental results, our method has achieved excellent performance across a rich set of benchmarks, which is sufficient to demonstrate the effectiveness of our approach.
>
> > [W2] The gating mechanisms use fixed thresholds... tuned for benchmarks but potentially suboptimal for diverse real-world scenarios.
>
> Thank you for this comment regarding our design methodology and hyperparameters. Our framework is founded on the clear principle of **query-awareness**, which posits that information filtering in streaming video should be guided by user intent, rather than by query-agnostic visual dynamics. We acknowledge this aspect was discussed in our limitations section, but we believe this configurability is a practical advantage. The use of adjustable hyperparameters provides significant operational flexibility, allowing the system's behavior to be tailored to specific application needs. For instance, a surveillance video might require the retention of more fine-grained information, necessitating a higher sensitivity to temporal changes. In contrast, a narrative video summary might only require focusing on major scene shifts, thus benefiting from a lower sensitivity to minor variations. Our framework allows for the configuration of appropriate thresholds to meet these differing requirements, providing explicit control points for task-specific adaptation. This stands in contrast to an end-to-end model where such sensitivity levels are implicitly learned and fixed after training, offering less adaptability at inference time.
>
> Furthermore, although these thresholds are static in our experiments, our method achieved excellent performance across multiple benchmarks, which demonstrates its versatility.
>
>
> > [W3] The RTAR ablation relied on simulation... due to unavailable official evaluation. Although response generation was temporally constrained, trigger detection had access to the full video.
>
> You have indeed noticed a key point, and we would like to clarify our evaluation process here. For our QueryStream model, this setup does not provide an unfair advantage. Our triggering logic is handled by a separate CLIP-like model, a process which only serves to generate patch masks and response signals; features from future frames are never used in the Video-LLM's comprehension. During inference at a trigger point t, the Video-LLM strictly receives only the frames up to time t with tokens pruned, ensuring no future information is accessed. Our architecture is capable of true frame-by-frame processing, and analyzing the full video at once is a simulation convenience that produces identical results.
>
> However, we acknowledge this setup's implications for the baseline. The open-sourced implementation of TimeChat-Online's triggering logic operates by processing the full video to identify the *k* points with the lowest token drop rates (e.g., top 10%) as response triggers. Since this process utilizes the main Video-LLM's vision encoder, it effectively allows the baseline model to "see" future video frames when determining its trigger points. Despite this potential advantage for the baseline, our model still demonstrated superior performance. We argue that achieving a better result even under these conditions more strongly validates the effectiveness of our proposed query-aware method.
>
> ---
> Thank you once again for your thoughtful and constructive reviews. We have carefully considered all your comments and will incorporate these changes into the revised version. We hope our response and planned modifications will resolve your issues, and we look forward to further discussions with you.

---

### Official Review · Reviewer_6ZdJ · 2025-11-01

**Soundness:** 2
**Presentation:** 4
**Contribution:** 1
**Rating:** 4
**Confidence:** 5

**Summary:**

This paper presents QueryStream, a framework for streaming video understanding that builds query-aware temporal representations to improve performance. The method dynamically fuses incoming frame features with past context, guided by query-conditioned similarity maps, allowing the model to emphasize temporally relevant segments as the video unfolds. The paper reports improvements on multiple benchmarks (streming + long video benchmarks) compared with existing streaming or offline VLMs.

**Strengths:**

- The introduction is very well written. It clearly defines the gap between offline video-language models and real-time streaming setups, motivating the need for a query-aware temporal design.

- The method is well organized, with a clear hierarchy between query-aware temporal modules, memory updates, and frame-level fusion.

**Weaknesses:**

- The method is presented as a new query-aware temporal representation, but in practice it relies on several manually tuned parameters and heuristic weighting functions between query and frame embeddings. The current mothod design appears quite sensitive to these hyperparameters even though they have discussed in A.4. Despite the hierarchical structure, the approach still feels heuristic and lacks a clear principle.

- The proposed framework mainly reorganizes existing components such as query-conditioned similarity and temporal fusion into a clean, well-structured pipeline. In the end, it looks more like a refined arrangement of standard similarity-based scoring rather than a fundamentally new approach, so the novelty feels limited.

- The paper emphasizes low-latency and lightweight design, but there is no analysis of runtime, throughput, or resource usage. In streaming scenarios, maintaining query-aware similarity maps and recurrent updates for each frame can be computationally heavy. Without latency or efficiency analysis, it is unclear whether the method is actually suitable for real-time use.

**Questions:**

- Why was OpenCLIP chosen as the encoder instead of more recent options like SigLIP that are already aligned with VLMs?

---

> ### Author Response · Authors · 2025-11-20
>
> Many thanks to your valuable comments and questions, which help us a lot to improve our work. We address your questions as follows.
>
> ---
>
> > [W1] The method is heuristic in practice, relying on tuned parameters and weighting functions. It is sensitive to these settings and lacks a clear principled foundation... despite its hierarchical structure.
>
> We thank the reviewer for this comment regarding our design methodology and hyperparameters. Our framework is founded on the clear principle of **query-awareness**, which posits that information filtering in streaming video should be guided by user intent, rather than by query-agnostic visual dynamics. The use of adjustable hyperparameters provides significant operational flexibility, allowing the system's behavior to be tailored to specific application needs. For instance, a surveillance video might require the retention of more fine-grained information, necessitating a higher sensitivity to temporal changes. In contrast, a narrative video summary might only require focusing on major scene shifts, thus benefiting from a lower sensitivity to minor variations. Our framework allows for the configuration of appropriate thresholds to meet these differing requirements, providing explicit control points for task-specific adaptation. This stands in contrast to an end-to-end model where such sensitivity levels are implicitly learned and fixed after training, offering less adaptability at inference time. We acknowledge this aspect was discussed in our limitations section, but we believe this configurability is a practical advantage.
>
> Furthermore, although these thresholds are static in our experiments, our method achieved excellent performance across multiple benchmarks, which demonstrates its versatility.
>
> > [W2] The framework reorganizes existing components into a clean pipeline. It appears more as a refined arrangement of standard similarity-based scoring rather than a fundamentally new approach, so novelty is limited.
>
> We appreciate your feedback and would like to further clarify the precise novelty of our work. We introduce a general, **training-free** framework that functions as a lightweight, plug-and-play module. Requiring only a standard CLIP-like model, QueryStream can **transform a Video-LLM into a proactive streaming video model**. with proactive response capabilities.
>
> This approach bridges a critical gap between two distinct classes of prior models. Proactive methods like TimeChat-Online are query-agnostic, operating on a "change-is-important" principle that conflates visual dynamics with semantic relevance, leading to inefficient processing and spurious responses. Conversely, query-sensitive methods like ReKV are entirely passive, unable to respond autonomously and relying on retrieving information from a historical KV-cache. Our framework, QueryStream, presents a more advanced solution. By operating directly on the incoming token stream, our QDP performs efficient, real-time filtering guided by user intent, unlike the query-agnostic approach of TimeChat-Online or the cache-retrieval mechanism of ReKV.
>
> > [W3] The paper emphasizes a lightweight design but lacks any analysis of runtime, latency, or resource usage.
>
> We thank you for pointing out this important omission. We agree that a quantitative analysis of computational efficiency is essential to substantiate our claims of real-time suitability.
>
> In practice, QueryStream's efficiency stems from its architecture. To obtain the pruning mask and response signal for each frame, our method requires only a single forward pass through a lightweight vision encoder (OpenCLIP-ViT-L/14) and a set of highly parallelizable vector operations. This process is significantly more efficient than approaches like TimeChat-Online, which must utilize the full, heavier vision encoder of the backbone Video-LLM for its change detection logic.
>
> To provide a more direct and intuitive comparison, we have conducted new experiments measuring both the decision-making overhead and the end-to-end inference latency. The results are shown in the table below.
>
> | Model | Token Keep Rate (%) | Decision Latency (ms/frame) | Inference Latency (ms/sample) |
> | :--- | :--- | :--- | :--- |
> | Qwen2.5VL (Full Token) | 100% | - | 3968 |
> | TimeChat-Online | 38.00% | 70.57ms | 2310 (×1.72 speedup) |
> | **QueryStream (Ours)** | **38.18%** | **47.01ms (×1.50 speedup)** | **2129 (×1.86 speedup)** |
>
> The table clearly demonstrates two key advantages of our approach. First, our decision-making process is **1.5x faster** per frame than TimeChat-Online's, confirming the benefit of our lightweight module. Second, this initial efficiency contributes to a superior **end-to-end speedup of 1.86x** over the full-token baseline, outperforming the baseline's own pruning method at a similar token keep rate. We will add this comprehensive efficiency analysis to the revised version of the paper to empirically verify the performance improvement.

---

> ### Author Response · Authors · 2025-11-20
>
> > [Q1] Why choose OpenCLIP over more recent, VLM-aligned encoders like SigLIP?
>
> In our framework, the CLIP-like model functions as an independent module for our QDP and RTAR mechanisms, generating pruning masks and response signals. It does not directly interact with the internal representations of the backbone Video-LLM. Therefore, whether the encoder is specifically "aligned" with the VLM is not a strict requirement for our method to function, although we agree that a more powerful encoder should generally yield better performance.
>
> To empirically validate the flexibility of our approach and explore the potential of more recent encoders, we followed your suggestion. Under the same evaluation setting described in Appendix A.3, we replaced our original OpenCLIP encoder with SigLIP and re-ran the experiment on the OVO-Bench subset, allowing a direct comparison with the OpenCLIP results reported in Table 6. The results are summarized below:
>
> | Feature Encoder | Score |
> | :--- | :--- |
> | OpenCLIP-ViT-L/14 | 48.5 |
> | **SigLIP-SO400M/14@384** | **48.7** |
>
> The results confirm that our framework seamlessly integrates with different encoders. While the more recent SigLIP encoder does provide a slight improvement, the moderate gain highlights that the core strength of our method lies in its query-aware logic. We will add these new results to the revised paper.
>
> ---
>
> Thank you once again for your thoughtful and constructive reviews. We have carefully considered all your comments and will incorporate these changes into the revised version. We hope our response and planned modifications will resolve your issues, and we look forward to further discussions with you.

---

> > ### Comment · Reviewer_6ZdJ · 2025-11-27
> >
> > The “query-awareness” caim does not seem persuasive because the method relies entirely on clip-like text encoders, which have very limited ability to represent open-ended or abstract queries. For example considering "summarize this video" or "is this action dangerous?", for such queries, the reviewer bielieves that the clip-like models cature semantic similarity primarily aligned with nouns and simple visual concepts, but they do not encode actual high-level user intention.
> >
> > So the reviewer thinks that “query-conditioned similarity maps” is somewhat primitive and cannot archieve indeed "query-aware temporal modeling" (tbh the reviewer thinks that the sensitivitiy on hyperparameters threshold is one of major weakness of this paper even if it achives higher performance than the baselines.)
> >
> > The reviewer carefully read other reviewers’ comments and the authors’ rebuttal, and thank you for the authors' efforts for further clarifications. However, the core concerns about the actual depth of “query-awareness” and the heavy reliance on heuristic thresholds still remains. Overall, while the framework is clearly written and empirically effective, the reviewer feels that the contribution is relatively weak and therefore leans slightly negative.

---

### Author Response · Authors · 2025-12-02

We thank the reviewers for their insightful feedback and constructive suggestions, which have been invaluable in helping us to refine and strengthen our paper. We believe we have thoroughly addressed all the major issues raised during the review process, providing detailed responses and clarifications. Furthermore, reviewer ioxx has raised the score to **8**.

Our contributions and strengths were acknowledged by the reviewers. They agreed that the paper presents a well-motivated and clearly articulated framework that shifts the paradigm to a more effective query-centric approach (**6ZdJ, ioxx**). A key highlight was our methodological innovation; reviewers found our proposed components, including Query-Aware Differential Pruning (QDP) and Relevance-Triggered Active Response (RTAR), to be novel, intuitive, and effective (**vYGP, ioxx**). Furthermore, they recognized that our framework is lightweight, training-free, and plug-and-play, making it suitable for real-time applications (**vYGP, UzVR, ioxx**). Our strong experimental validation, which establishes new state-of-the-art results on benchmarks like StreamingBench and OVO-Bench with significantly fewer tokens, was also appreciated (**UzVR, ioxx**).

We summarize and clarify some common questions raised by reviewers here:

1.  **A Principled and Novel Framework for Proactive Video Understanding:** Our primary contribution is a novel framework founded on the clear principle of **query-awareness**, which bridges a critical gap in prior work. It advances beyond proactive, query-agnostic models (e.g., TimeChat-Online) by aligning processing with user intent, and beyond passive, query-sensitive models (e.g., ReKV) by enabling autonomous, proactive responses. Our design is a **general, training-free, plug-and-play** module that effectively transforms any standard Video-LLM into an intelligent streaming agent. This was clarified in our detailed responses to Reviewers **6ZdJ** and **ioxx**.

2.  **Robustness, Flexibility, and Generalizability:** We have addressed all concerns regarding the robustness and generalizability of our approach. Our framework demonstrates significant **operational flexibility**: its hyperparameters can be adjusted to meet the specific demands of different tasks and scenarios, such as prioritizing fine-grained detail in surveillance or major events in summarization. This configurability is a practical advantage over fixed-behavior models. Crucially, our architecture also inherently supports **multi-turn dialogue** by dynamically updating the query embedding. In terms of generalization, we have already demonstrated our module's effectiveness when integrated with multiple diverse backbones, including **Qwen2.5-VL, InternVL2.5, and InternVideo2.5** (as shown in our Appendix). Furthermore, to address reviewer feedback:
    *   **Encoder Flexibility:** We successfully integrated a more recent encoder **(SigLIP)**, demonstrating that our framework is model-agnostic and benefits from ongoing advancements (response to **6ZdJ, vYGP, UzVR**).
    *   **Backbone Model Generalization:** We have committed to adding results on an **additional backbone Video-LLM** in the final version to further strengthen this claim (response to **ioxx**).

3.  **Comprehensive Efficiency Validation:** In response to feedback from Reviewers **6ZdJ** and **ioxx**, we have conducted new, detailed experiments to **quantify the efficiency** of our framework. The new results provide concrete evidence of our module's low latency and significant end-to-end inference speedup (e.g., **1.86x** over the baseline), empirically validating our claims of real-time suitability.

We are carefully addressing all feedback and are confident that these clarifications and new results have addressed the reviewers' primary concerns. We will incorporate these updates into the camera-ready version.

---

### Meta-Review · Area_Chair_QUBG · 2026-01-02

**Summary:**

This paper proposes QueryStream, a training-free framework for streaming video understanding that introduces query-aware pruning and proactive response scheduling. Reviewers recognize the work as well-motivated and practically relevant, with a clear shift from query-agnostic change detection to intent-driven processing. While some reviewers initially question the heuristic nature of the design and the reliance on fixed thresholds, the rebuttal provides additional efficiency analysis, encoder generalization results, and clarifications that address the main concerns. After reviewing the revised evidence and discussion, the AC recommends acceptance.

**Reviewer Concerns:**

Reviewer concerns mainly focus on the principled nature of the query-aware design, sensitivity to hyperparameters, and the lack of initial efficiency analysis. These concerns are addressed through added latency and throughput measurements, encoder ablations (including SigLIP), and clearer justification of the query-aware formulation and response mechanism. While the method remains logic-based rather than learned end-to-end, reviewers generally agree that the design is effective, interpretable, and suitable for streaming scenarios. No major concerns remain that would block acceptance.

**Reviewer Scores:**

- Reviewer 6ZdJ: Initially assigns a score of 4, expressing concerns about heuristic design and efficiency claims. After the rebuttal and added runtime analysis, the assessment would remain borderline reject (4).
- Reviewer vYGP: Initially assigns a score of 4, noting questions about encoder limitations and threshold sensitivity. Following the rebuttal clarifications and additional experiments, the assessment would remain slightly positive (6).
- Reviewer UzVR: Initially assigns a score of 6, recognizing the practical value and clarity of the framework. After the rebuttal addressing efficiency and robustness, the assessment would remain positive (6).
- Reviewer ioxx: Initially assigns a score of 6. After the rebuttal addressing concerns about novelty, efficiency, and generalization, the assessment would remain positive (6).

---

### Decision · Program_Chairs · 2026-01-26

Accept (Poster)